# Spatiotemporal changes in river network connectivity in the Nile River Basin due to hydropower dams

Anthony Basooma[1,¤a,¤b,☺,*], Astrid Schmidt-Kloiber[1], Rose Basooma[2,3], Herbert Nakiyende[4,¤a], Johannes Kowal[1,5,6], Andrea Funk[1,5,6], Thomas Hein[1,5,6], Florian Borgwardt[1,5,6,☺,*]

1 BOKU University, Institute of Hydrobiology and Aquatic Ecosystem Management, Vienna, Austria, 2 BOKU University, Institute for Integrative Nature Conservation Research, Vienna, Austria, 3 Department of Natural Resources Economics, Faculty of Natural Resources and Environmental Sciences, Namasagali Campus, Busitema University, Namasagali, Uganda, 4 Department of Zoology, Entomology and Fisheries Sciences, College of Natural Sciences, Makerere University Kampala, Kampala, Uganda, 5 Christian Doppler Laboratory for Meta Ecosystem Dynamics in Riverine Landscapes, University of Natural Resources and Life Sciences, Vienna, Austria, 6 Department Water-Atmosphere-Environment, Institute of Hydrobiology and Aquatic Ecosystem Management, Vienna, Austria

☺ These authors contributed equally to this work.
¤aCapture Fisheries and Biodiversity Conservation, National Fisheries Resources Research Institute, Jinja, Uganda
¤bBiodiversity Analytics and Ecohydrology, Asterlook International Limited, Jinja, Uganda
* anthony.basooma@boku.ac.at (AB), florian.borgwardt@boku.ac.at (FB)

## Abstract

Dams offer indispensable services, including hydropower generation, water for irrigation, and flood mitigation. These barriers disrupt natural river flows, affecting river ecosystems structural and functional connectivity. The number of hydropower dams in the Nile River Basin has increased during the last decades. However, little is known regarding the spatiotemporal variation in the river network fragmentation and the individual dam contributions. We used the Reach Connectivity Index (RCI) and network centrality measures to assess temporal and spatial changes in longitudinal connectivity. We selected the Equatorial Nile and Blue Nile basins, the major hotspots for present and future hydropower developments. We collated 101 existing, under construction, and proposed dams in the Equatorial Nile Basin (ENB) from 1954 to 2035 and 19 dams in the Blue Nile Basin (BNB) from 1925 to 2035. The dams in the ENB have significantly increased over time, with a sharp increase recorded between 2010 and 2015. The mean RCI for the BNB, considering impassable dam scenarios, ranged from 62.5% (SD = 12.5%) in 1925 to 21.35% (11.6%) in 2035. Similarly, in the ENB, the mean RCI for impassable dam scenarios ranged from 50.1% (SD = 2%) in 1954 to 18.1 (12.1%) in 2035. River segments in the middle section of both basins were the most affected. The decline in the mean RCI was significantly higher in the ENB compared to BNB. The reduced connectivity is mainly in the ENB, threatening the basin's biodiversity. Existing dams, including the Grand Ethiopian

**Data availability statement:** Data used in Figures 1,5 and 6 are reprinted from [https://www.hydrosheds.org/hydroatlas] under a CC BY license, with permission from [HydroSHEDS], original copyright [2022]. The checked datasets for the dams used for the Blue Nile Basin and Equatorial Nile Basin have been archived on Figshare for easy accessibility (https://doi.org/10.6084/m9.figshare.26886409.v1)

**Funding:** Anthony Basooma acknowledges funding from AquaINFRA (agreement ID: 101094434) and DANUBE4All (grant agreement no. 101093985) projects funded by the European Commission. The financial support by the HR21 Doctoral School of the University of Natural Resources and Life Sciences, Vienna, Austria, the Austrian Federal Ministry for Digital and Economic Affairs, the National Foundation for Research, Technology, and Development, and the Christian Doppler Research Association is gratefully acknowledged. The funders had no role in study design, data collection and analysis, decision to publish, or preparation of the manuscript.

**Competing interests:** The authors have declared that no competing interests exist.

Renaissance Dam on the BNB and Kakono on the ENB, should have effective fish migratory corridors that allow the passage of fish either upstream or downstream. We also recommend establishing a detailed basin-wide database for barriers and assessing their passability to understand the full extent of the river network fragmentation. We also recommend regular monitoring of barrier impacts by integrating safe, cost-effective methods such as remote sensing and environmental DNA (eDNA) to assess both flora (macrophytes, phytoplankton) and fauna (macroinvertebrates, fish, zooplankton).

## 1. Introduction

The growing demand for energy and flood control has led to increased construction of barriers along rivers [1–3]. These barriers disrupt natural river flows and their structural and functional connectivity [4–7]. Disrupted flow regimes are coupled with sediment deficits and altered nutrient flow in the downstream sections [8–10]. The barriers also affect river geomorphology [9,11,12] and species migration routes to critical habitats such as spawning and feeding grounds [8,9,13]. The fragmented river sections restrict species movement and form small population patches, reducing gene flows and adequate population size within isolated ecosystems [14–16].

Due to high dependence on migratory corridors, migratory fish species are more threatened. [13,17]. Their populations have declined by 81% globally since the 1970s [18,19], compared to 69% for all wildlife animal populations [20]. The decline in migratory fish species coincides with the 10-fold increase in large dams from 1950 to 2017 [21]. These dams are constructed even in protected areas [21,22], biodiversity hotspots vital for freshwater ecosystem conservation [23]. Most large rivers (>1,000km of river length) are fragmented [24], and the non-fragmented rivers are also facing threats from future dam constructions [25].

Globally, river network connectivity analysis has emerged as an essential tool for assessing the ecological health of rivers [26]. The degree of river fragmentation is usually evaluated using network connectivity indices based on graph theory principles [27,28]. Also, stream fragmentation metrics are used to evaluate the extent of fragmentation due to dams [29]. The scale of analysis varies from the global extent [3,4] to the local catchment scale [30]. Although global-scale analysis may not be used to define the individual barrier contribution to river network fragmentation, it is vital in identifying basins or regions that are comparably fragmented [5]. In contrast, local or catchment scale analysis is beneficial in evaluating individual dam contributions to fragmentation, but the outputs may not be extrapolated to other river basins. This is due to differences in the structural configuration of the river networks, the number of barriers, the size of the basin, and functional components like the ability of specific fish species to overcome obstacles. Therefore, river basin-specific analyses should be conducted to deepen the understanding of the connectivity issues.

Notwithstanding the cosmopolitan nature of river fragmentation [5,31], river basin-specific network connectivity analyses at a catchment scale are few in the

Global South [32]. For instance, despite the Nile River's economic and ecological significance, its network fragmentation extent is still less understood. The Nile River connects extended wetlands and the Equatorial Lakes regions, including Lake Victoria and Lake Kyoga, which provide habitats for diverse aquatic fauna [33,34]. Hydrologically, the river provides ecosystem services to about 280 million people in eleven countries across different cultures, communities, and landscapes [34]. The Nile is an umbilical cord supporting Sudan and Egypt as the primary water source for industrial, irrigation, and agricultural purposes [35,36]. The river deposits nutrient-enriched sediments in the downstream flood plains, driving agricultural production and economic development in the riparian countries [37]. Due to this strong dependence on the river resources, hydro-political conflicts over user rights have already spurred for the last century among the basin countries [35,37,38]. Diplomatic escalations have emanated due to the Grand Ethiopian Renaissance Dam (GERD) construction within the Blue Nile subbasin [36,37,39]. Also, numerous new dams are being proposed or are under construction [40]

Hydropower developments within the Nile River Basin are mainly concentrated in the Equatorial Nile and the Blue Nile subbasins [40]. The subbasins are the primary source of water discharge, feeding the basin downstream, with the Blue Nile subbasin contributing about 60% of the total water budget in the Nile during the rainy season [34]. Therefore, this study assessed the longitudinal connectivity of the Equatorial Nile and Blue Nile subbasin. Specifically, we examined the temporal trends in longitudinal connectivity and spatial variation in fragmentation patterns of the subbasins due to hydropower dams. Because the number of barriers and their location determines river fragmentation [41], we examined spatial variations in river segments due to dam construction along the river network. We also examined the individual dam contribution to river fragmentation, which can be incorporated into barrier removal prioritization metrics. We hypothesized that: (1) there is a significant temporal increase in the number of dams constructed on the river segments in each subbasin. (2) There is a significant relationship between longitudinal connectivity and the position of river segments in the network, and (3) individual dams have varying contributions to the reduction in river longitudinal connectivity.

## 2. Materials and methods

### 2.1. Investigation area and dam data

The Equatorial Nile Basin (ENB) consists of a section of the Upper White Nile (234,680 km²) and Equatorial Lakes (394,147 km²) [42] (Fig 1). The Blue Nile Basin (BNB) is located on the eastern side of the Nile River Basin (NRB) and covers about 350,000 km² at the confluence in Sudan [33]. The BNB emanates from the highlands in Ethiopia and flows downstream, joining the White Nile in Sudan, and receives water from Lake Tana [35].

The two subbasins were extracted from the global BasinATLAS_v10_lev4 [42] and the river network was clipped from the global RiverATLAS [42]. The basin and river network layers were processed in Q-GIS version 3.28.4 [43]. The global basin and river networks layer were retrieved from (https://figshare.com/ndownloader/files/20087237) and (https://figshare.com/ndownloader/files/20087321), respectively [42]. The resultant basin and river network layers were archived in Figshare (accessible via https://doi.org/10.6084/m9.figshare.26886409.v1).

Existing, proposed, and dams under construction were collated from the Renewable Power Plant Database for Africa (RePP Africa) (an open-access dataset available at https://doi.org/10.6084/m9.figshare.c.6058565.v1). Additional information on the date of commissioning was collated from the literature [44,45]. The updated dam list is archived at (https://doi.org/10.6084/m9.figshare.26886409.v1). Hydropower dams with commissioning or construction dates were retained in the analysis to account for temporal changes in river network connectivity.

### 2.2. Assessing river network connectivity

We included all river segments of all stream orders in the temporal and spatial network connectivity analysis. Most studies ignore the low-order stream networks (i.e., especially segments of stream orders 1–3). We obtained the river network,

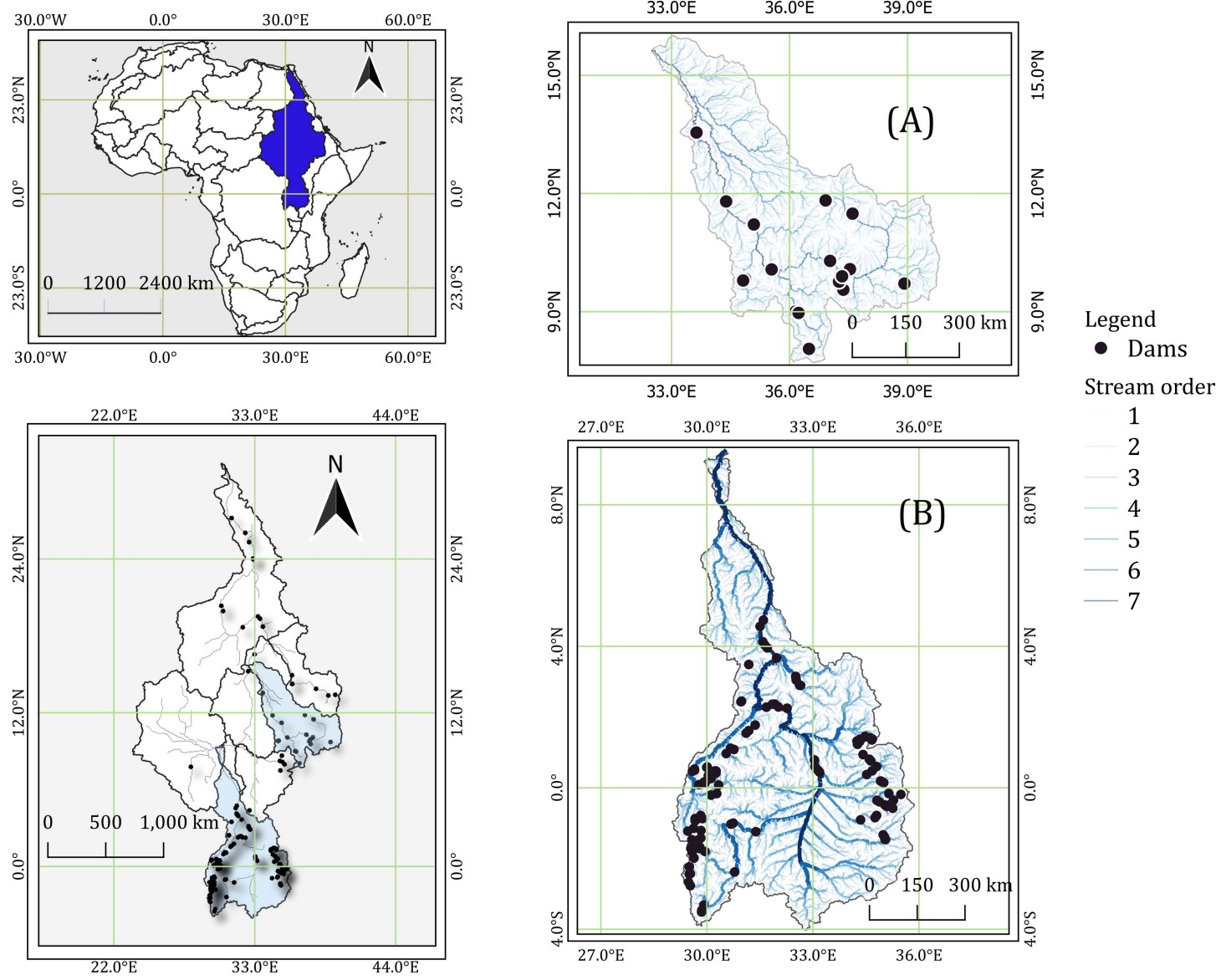

**Fig 1. Overview of the investigation area: The Blue Nile Basin (BNB) (A) and Equatorial Nile Basin (ENB) (B) from the Nile River Basin (NRB).**
Maps on the left show the location of the Nile River Basin within Africa (upper left) and the location of BNB and ENB in the NRB (lower left). The basin layers were extracted from RiverATLAS and BasinATLAS_v10_lev4 [42]. The points indicate dam locations extracted from [40,44,45]. Reprinted from [https://www.hydrosheds.org/hydroatlas] under a CC BY license, with permission from [HydroSHEDS], original copyright [2022].

which contained information for each river segment on length, stream order, upstream catchment area, and flow order [42]. We identified artificial barriers within the basin and ignored natural barriers like waterfalls. Hydropower dams were the only artificial barriers considered in this study due to available data, most especially the date of commissioning [40]. The dams were snapped to the river network with a distance threshold of 1 km, and the correct position of the snapped barrier was to ensure an alignment with the river segments.

Barrier passability measures the probability of an organism passing via a barrier from either downstream or upstream (King et al., 2017). It varies based on the species life history stages (adult or juvenile), swimming abilities, height/size, and

barrier material used [46]. However, a detailed quantification of each barrier's upstream and downstream passability was impossible due to unavailable data on mostly dam passibilities. Thus, we chose a scenario-driven approach where different varying barrier passabilities were considered and scenarios were assessed. In these scenarios, the upstream and downstream passability increased by 0.1 steps from 0.0 to 0.9 (0.0 indicated impassable dams, and 0.9 indicated highly passable dams).

## 2.3. Computing river network connectivity

We examined river network connectivity using the Reach Connectivity Index (RCI) for potamodromous species. RCI was based on connected river segments that allow species to migrate upstream or downstream without limiting species dispersal [47]. The probability of realizing structural connectivity for a river reach for potamodromous species was based on the fraction of the reach length ($l_i$) relative to the total length ($L$) of the river network (Eq 1). No species was considered since the index parameters are not species-specific [48].

$$RCI = \sum_{i=1}^{n} lij\frac{l_i}{L} * 100$$

(1)

Where n is the number of river segments equal to the total number of barriers and confluences plus 1 [46]. RCI of 0 indicates a fully isolated segment, and 100 indicates a segment fully connected to other segments. Since no individual species were considered, only the structural component ($C_{ij}$) of the dispersal probability ($I_{ij}$) was extracted (Eq 2). The barrier passability determined the structural component $C_{ij}$ (Eq 3).

$$I_{ij} = C_{ij} * B_{ij}$$

(2)

The dispersal component ($B_{ij}$) was dropped, and $C_{ij}$ was computed according to Eq 3.

$$c_{ij} = \prod_{m=1}^{M} P_m^2$$

(3)

Where $p_m$ equals the passability of the m$^{th}$ dam along the path from the segment $i$ to $j$.

## 2.4. Temporal changes in the river network connectivity

We evaluated the changes in the mean RCI since the first dam was commissioned for each basin. The first dam was commissioned on the Blue Nile in 1925 and 1954 for the ENB [40]. To account for future changes in network connectivity, we considered proposed dams or those currently under construction based on the date of commissioning in available literature [44,45]. Data checks were conducted to determine dams that fall outside the basin spatially. Also, the year of first commissioning or the start of construction was checked.

We calculated the mean RCI values over ten years from when the first dam was commissioned to 2035. We classified the RCI values based on the dam passability scenarios as i) impassable (with passability of 0), ii) lowly passable (0.1 to 0.3), iii) moderately passable (>0.3 to 0.6), and (iv) highly passable (>0.6 to 0.9) [46]. Differences in the mean RCI among the passability categories and across the periods were assessed using a Two-way ANOVA. We conducted a post-hoc Tukey test to determine whether group pairwise differences differed significantly. We set a threshold of 50% to assess management intervention about the reduction in connectivity. The student's t-test with a known mean was used to test if the mean RCI had significantly reached the threshold level.

 

## 2.5. Spatial variation in the network connectivity

Besides the RCI, for each segment, further centrality measures, including degree centrality (number of nodes connected to a particular node), betweenness centrality (shortest distance to other nodes), closeness centrality (a measure of how close the node is to other nodes), and bonacich centrality (a node connected to other highly connected nodes), were computed (Rodrigues, 2019). The degree, betweenness, closeness, and power_centrality from the igraph package were used, respectively [49]. Centrality metrics were compared with RCI calculated for the 2035 period, assuming an upstream and downstream barrier passability of 0.5. This passability was selected because it is an average of all investigated passabilities. We used the 2035 time period because it includes all dams and represents conditions that can realistically be expected as the future status quo for the basins. The Pearson correlation coefficient was used to determine the relationship between different centrality measures and the RCI in each basin. Principal Component Analysis (PCA) examined the relationship between centrality measures, mean RCI, and individual dams. The PCA was computed using the prcomp function (stats) and visualized with the fviz_pca_biplot function (factoextra) [50].

## 2.6. Dam contribution on the river network fragmentation

We determined the contribution of each dam to the fragmentation of the whole river network using the leave-one-out principle (Eq 4) [51,52]. The barrier contribution to river fragmentation is vital in barrier removal prioritization and financial and ecological cost-effectiveness [41,53].

$$dCCI = \left( \frac{CCI_{start}\, m - CCI_{start}}{CCI_{start}} \right)$$

(4)

Where $dCCI$ is the change in the whole catchment connectivity due to the fragmentation of river segments. $CCI_{start}\, m$ is the change in the catchment connectivity when a particular dam $m$ is left out [51]. $CCI_{start}$ is the connectivity when all dams are considered. The higher the $dCCI$, the higher the contribution of a particular dam to fragmentation.

## 3. Results

### 3.1. Temporal changes in the basin river network connectivity and variation from the thresholds

We collated 19 dams in the BNB from 1925 to 2035. Eight dams were already constructed; one was under construction, and ten were proposed. The number of already constructed and proposed dams has steadily increased since 1925 (Fig 2). Correspondingly, the mean RCI decreased significantly in 2015 (Fig 3). Evaluating different dam passability scenarios showed a significant decrease in RCI from 55.6% (SD=14.7%) in 1925 to 19.1% (SD=12.3) in 2035 for the impassable scenario (Fig 2). For the lowly passable scenario, RCI decreased from 62.2% (SD=12.7) in 1925 to 21.4% (SD=11.6) in 2035. In the moderately passable scenario, RCI decreased from 73.3% (SD=9.6) in 1925 to 27.9% (SD=10) in 2035. Lastly, for the highly passable scenario, RCI decreased from 88.9% (SD=6) in 1925 to 54% (SD=14.7) (Fig 2). In summary, based on the lowest mean of the RCI observed in the impassable dam scenario and the highest mean of the highly passable scenario, the basin network connectivity ranged from 19.1% to 88.9%.

Regarding the BNB, the two-way ANOVA showed significant differences among the passability scenarios ($F_{(3, 1794205)}$ = 726933, p<0.01) and periods of dam construction ($F_{(11, 1794205)}$ = 145964, p<0.01). The Post-hoc Tukey test showed pairwise significant differences among most passability scenarios and periods of dam construction after 1955, but also 1965, 1975, and 1985. For the impassable dam scenario, the mean RCI was significantly lower than 50% (the management threshold) by 1965. Regarding the lowly passable scenario, the threshold was reached between 1985 and 1995 and between 2015 and 2025 regarding the moderately passable scenario (Fig 1). However, in the high passable dam scenario, the mean RCI did not reach the threshold of 50%. (Fig 3).

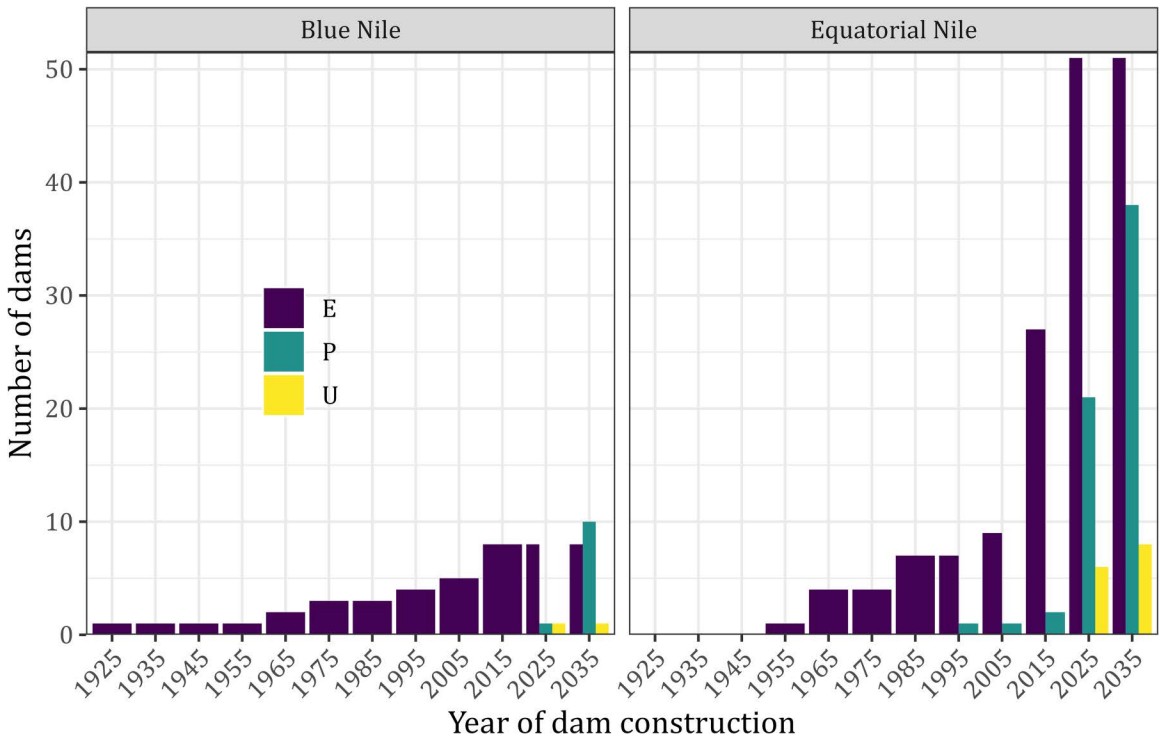

**Fig 2. Trends in dam construction during different periods for Blue Nile Basin (BNB) and Equatorial Nile Basin (ENB).** The first dam construction was reported on the BNB in 1925 compared to 1954 on ENB. However, a significant increase in the dams proposed for construction and already under construction was recorded in the ENB, with a noticeable surge after 2010.

In the ENB, we collated 129 dams from 1955 to 2035. Two dams were under construction; 45 were existing, 44 were proposed, and 28 had no proposed construction dates. The number of dams substantially increased between 2010 and 2015 (Fig 2). For the impassable dam scenario, the mean RCI decreased from 50.1% (SD=2) in 1955 to 18.2% (SD=12.9) in 2035. Regarding the low passable scenario, the mean RCI decreased from 57.5% (SD = 3.1) by 1955 to 18.9% (SD=12.71) in 2035. When considering the moderately passable barriers, the mean RCI decreased from 70% (SD=4.3) in 1955 to 21.0% (SD=12.2) in 2035. Finally, in the highly passable scenario, the mean RCI decreased from 87.5% (SD=5.6) by 1955 to 36.57% (SD=15.5). In summary, based on the lowest mean RCI of the impassable scenario and the highest mean when high passability was assumed, the RCI ranged from 18.2% to 87.5%.

Two-way ANOVA showed significant differences in the mean RCI across the passability scenarios ($F_{(3, 2366194)}$ = 1464347, p<0.01) and periods of dam construction ($F_{(8, 2366194)}$ = 1108539, p<0.01) (Fig 2). In contrast to the BNB, the mean RCI was already significantly similar to the threshold of 50% by 1955, after only one dam was constructed in the impassable dam scenario. Also, in the lowly passable scenario, the threshold was reached by 2005, and the moderately passable scenario by 2015. In contrast to the BNB, the mean RCI reached the threshold even in the highly passable scenario between 2025 and 2035 (Fig 3).

### 3.2. Spatial network connectivity

The spatial arrangement of dams influenced the mean RCI for each river network. For individual dams, the PC1 and PC2 explained 65.5% of the variations in the individual river reach centrality measures and RCI for both basins (Fig 4). Notably,

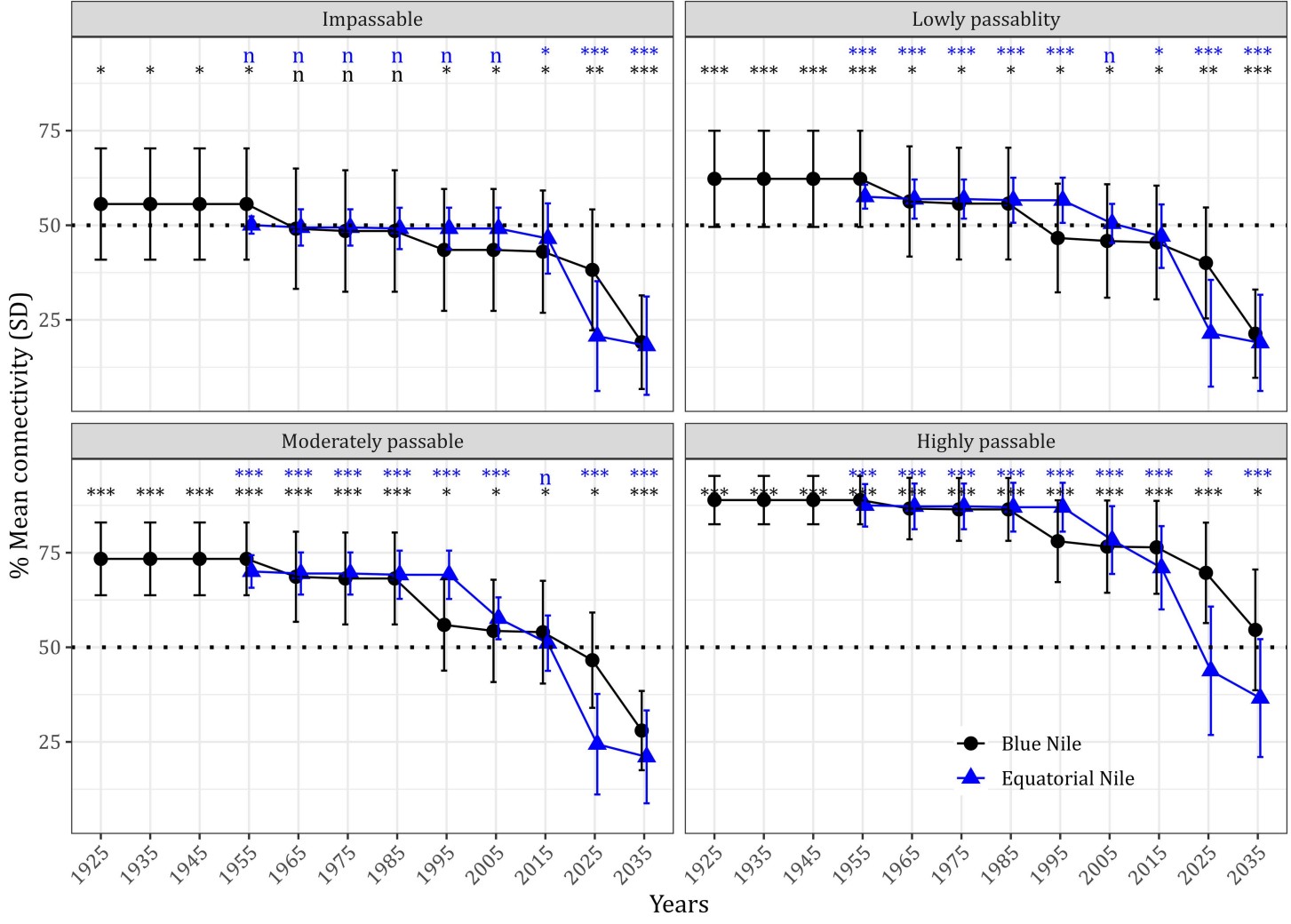

**Fig 3. Trends in the reach connectivity index for the Blue Nile Basin (black line) and Equatorial Nile Basin (blue line) under varying dam passability scenarios.** The horizontal black dotted line is the threshold when the connectivity reduces to 50% of the natural connectivity without dams on the network. *: 0.01 **: 0.001, and ns: not significantly different from the threshold.

the PCA showed that the Grand Ethiopian Renaissance Dam (GERD) (ID = 1) in the BNB was isolated from other dams, indicated by closeness and betweenness centrality (Fig 4). Moreover, the betweenness centrality of GERD (14.6) was the highest in BNB, followed by the dams Mendya (14.5), Roseires (14.4) and Karodobi (14.2).

Graphically, the northeastern section of the BNB had low centrality measures but high RCI (Fig 5). The river segments in the southwestern and central-eastern parts of the basin had the lowest connectivity. Generally, the betweenness centrality was similar to the RCI river network fragmentation but not for closeness and bonacich centrality. The degree of centrality was similar for most nodes (Fig 5). Negative bonacich centrality values were observed in the central-west part of BNB, which also had the highest closeness centrality (Fig 5).

For the ENB, the river segments in the south-central region had the highest RCI, followed by the northmost regions (Fig 6). The southwestern regions and patches in the central section had the lowest RCI. Betweenness also had a similar pattern to RCI but not closeness, degree, and bonacich, as observed in BNB.

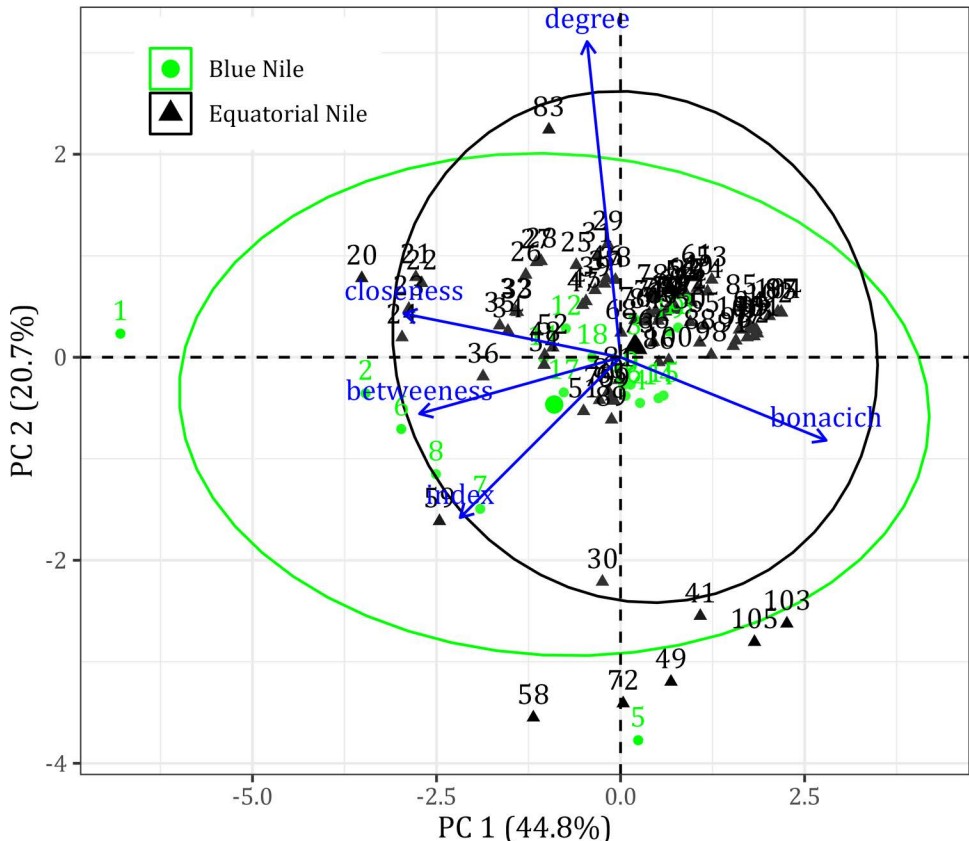

**Fig 4. Principal component analysis of the network connectivity index and network centrality measures for the Blue Nile and Equatorial Nile Basins.**

The relationship between RCI for river segments and centrality measures showed that degree centrality had a weak correlation with other centrality measures and RCI (Fig 7). In the BNB, RCI had a strong positive correlation with betweenness centrality and a moderate positive correlation with closeness centrality. In contrast, bonacich centrality had a moderately negative correlation with RCI (Fig 7). Among the centrality variables, bonacich centrality had a strong negative correlation with closeness centrality but moderately negative for betweenness centrality. Closeness centrality had a moderately positive correlation with betweenness centrality (Fig 7). In contrast to BNB, betweenness centrality had a moderate positive correlation with RCI (Fig 7). However, for the centrality measures, the correlations were similar to BNB.

### 3.3. Individual dam contribution to river network fragmentation

Generally, most dams had a low impact on the overall network connectivity in the BNB and the ENB. In the BNB, based on the change in the connectivity after using the leave-one-out principle, the dam contribution to network fragmentation was dominated by the Karodobi dam, followed by the dams Sennar and Mendya (Fig 8a). In the ENB, the Nabuyole dam planned at the River Nzoia will significantly contribute to the fragmentation of the river network by 2035, followed by the Fula Small dam and the Kakono dam (Fig 8b).

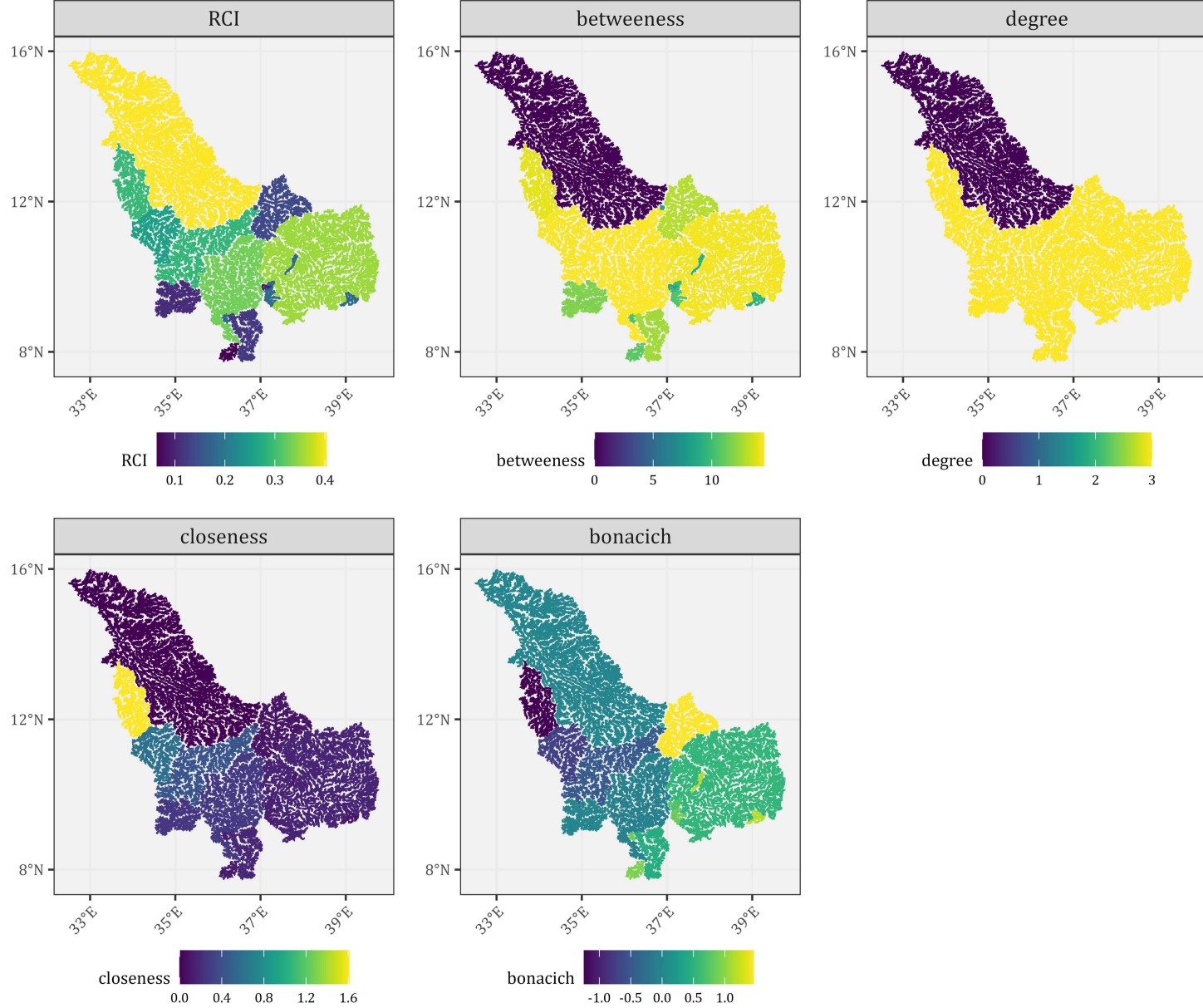

**Fig 5. The network centrality measures and connectivity scores of the Blue Nile Basin.** A dam passability of 0.5 was used, and river segments of all stream orders (1 to 7) and all dams were considered. Reprinted from [https://www.hydrosheds.org/hydroatlas] under a CC BY license, with permission from [HydroSHEDS], original copyright [2022].

## 4. Discussion

### 4.1. Temporal trends of connectivity in the basins

We thoroughly examined the temporal and spatial patterns in the river network connectivity of the Equatorial Nile and Blue Nile, focusing on individual dam contributions to overall river network fragmentation. In all passability scenarios, the RCI was lowest in the ENB compared to the BNB. Also, considering the 50% threshold, the trends in the mean RCI of the ENB reached the threshold even in a high passability scenario. Further, in both basins, a significant decline in connectivity

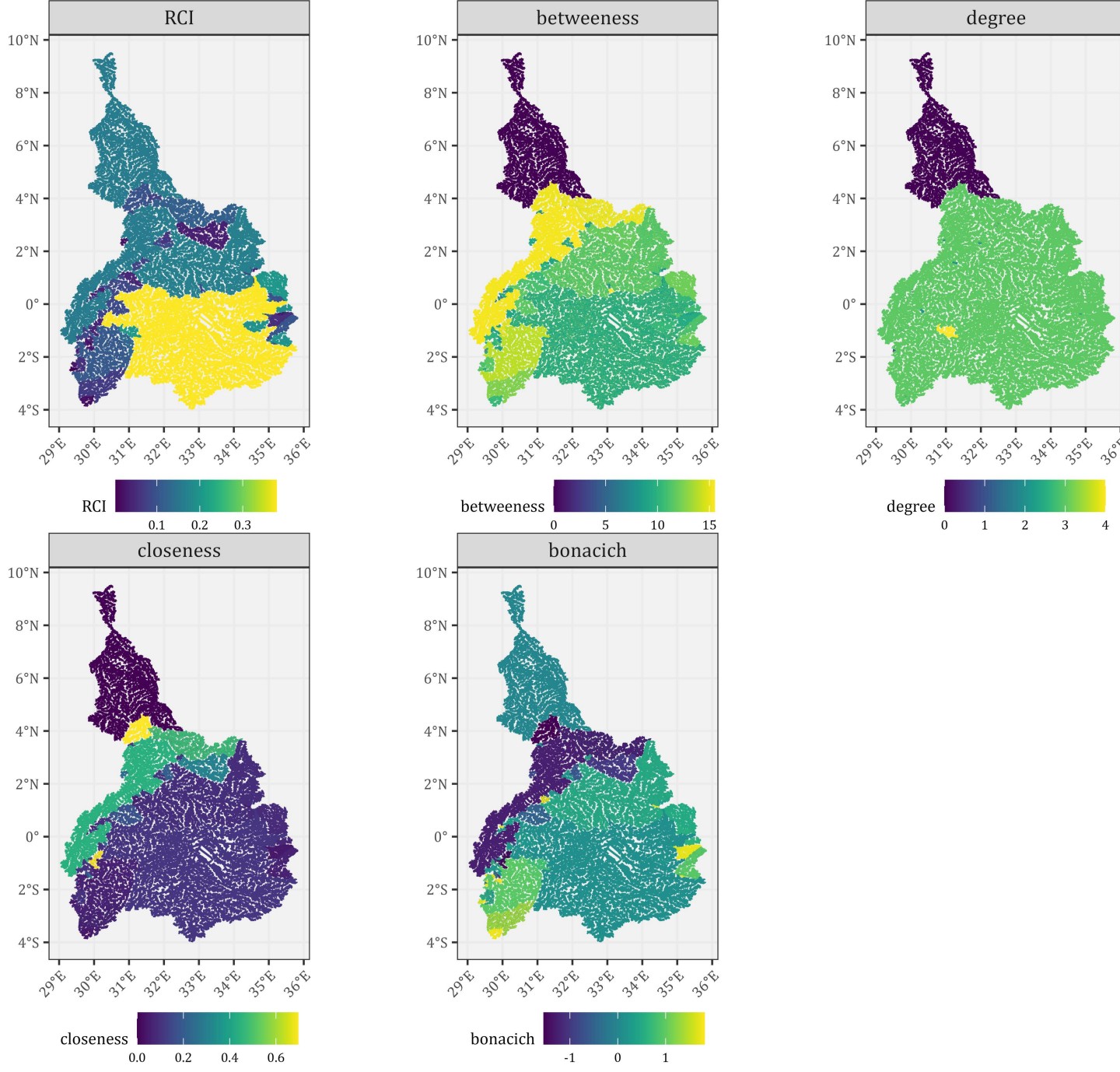

**Fig 6. The network centrality and RCI of the Equatorial Nile Basin.** A dam passability of 0.5 was used, and river segments of all stream orders (1 to 7) and all dams were considered. Reprinted from [https://www.hydrosheds.org/hydroatlas] under a CC BY license, with permission from [HydroSHEDS], original copyright [2022].

was recorded after 2010. The temporal trends in hydropower dam construction and the longitudinal connectivity changes observed in our study are highly linked with the Nile water acquired rights through colonial treaties and agreements [54]. For instance, Egypt acquired exclusive rights on any irrigation or hydropower dam projects to be constructed along the

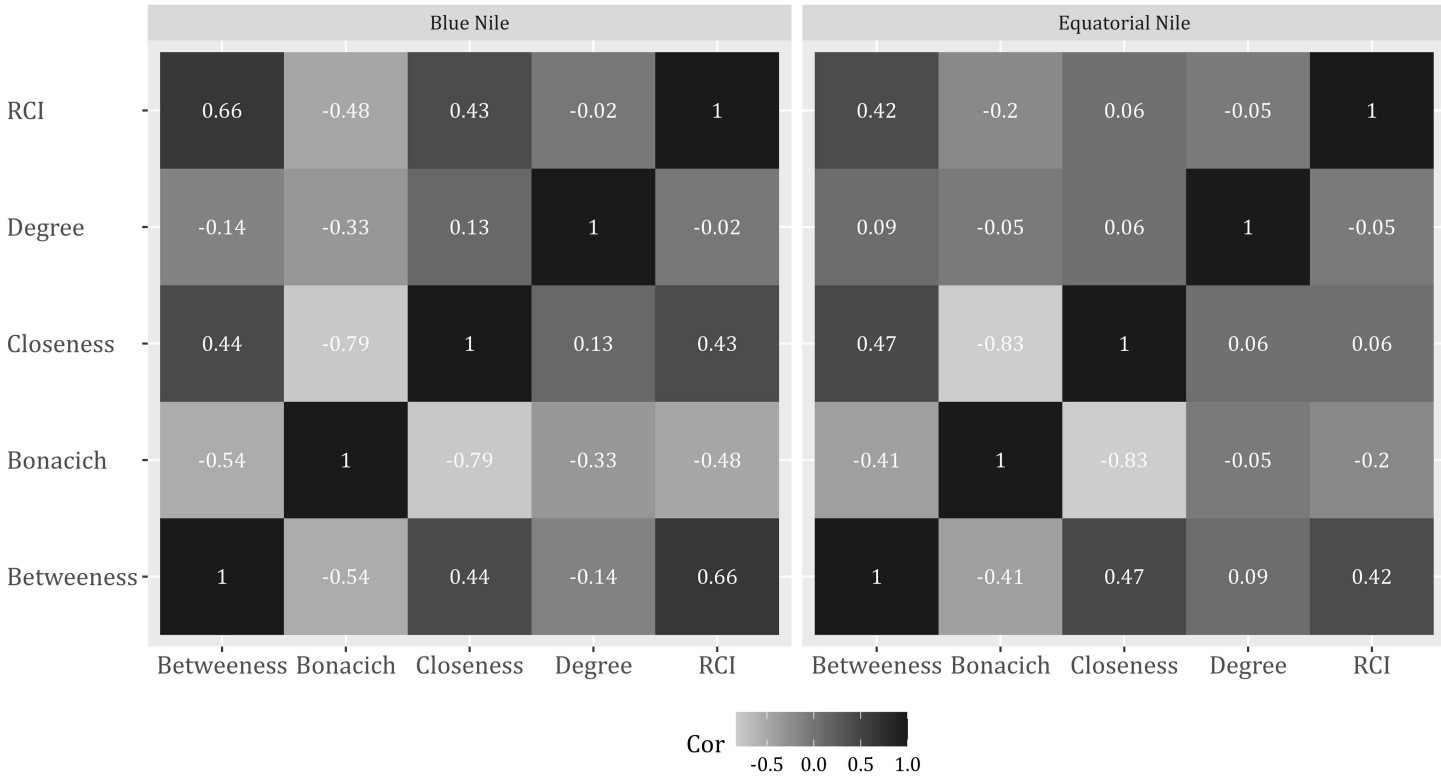

**Fig 7. The correlation coefficient among centrality measures and RCI for both basins.**

Nile in 1929 [37,54,55]. Consequently, in 1954, Egypt approved the construction of the Owen Falls dam in Uganda [56]. These rights were further reechoed in the 1959 Nile Treaty, bilaterally signed by Sudan and Egypt, which allocated 66% of the Nile waters to Egypt, 22% to Sudan, and 12% for natural losses such as evapotranspiration [54]. The 1959 Treaty enabled the construction of the Roseires Dam in the BNB, which was completed in 1966 [56]. These colonial agreements and the lack of technical capacity have regulated dam developments within the Nile River Basin [37]. However, in May 2010, the Nile Cooperation Framework Agreement (NCFA) was signed by the upstream countries, including Rwanda, Uganda, Tanzania, Kenya, and Burundi [55]. The agreement shrunk the veto rights of Egypt and introduced the no-harm principle, which involves conducting activities that do not harm the water flow to the downstream countries [55]. Subsequently, after 2010, we observed a significant increase in hydropower developments in the ENB, although no significant increase in the BNB. For instance, out of the 49 dams existing in the ENB, 25 (55.6%) were commissioned after 2013 [40]. In the BNB, fewer dams were commissioned. However, the Grand Ethiopian Renaissance Dam (GERD) has already caused diplomatic tensions between Ethiopia and Egypt and may lapse into a military conflict [36].

The trend in the fragmentation in the ENB was comparably higher in the BNB, which requires effective subbasin and Nile basin-wide planning. For example, several large dams, including Bujagali Hydropower Dam, Nalubale, Isiimba, and Karuma HP, have already been constructed along the Upper Victoria Nile within ENB. These dams are closely spaced, which has led to overlap in biodiversity offset areas (Inspection Panel, 2020). Other large dams, including Oriang (395 MW), Fula (890 MW), and Ayago (840 MW), are proposed in the middle region of ENB [40], which will significantly fragment the main channel of the river network. The dams have led to changes in fish species stocks and genetic isolation [15,57]. The tributaries, including Kagera, Mpanga, and Nzoia, are also heavily dammed, probably affecting the species migration routes.

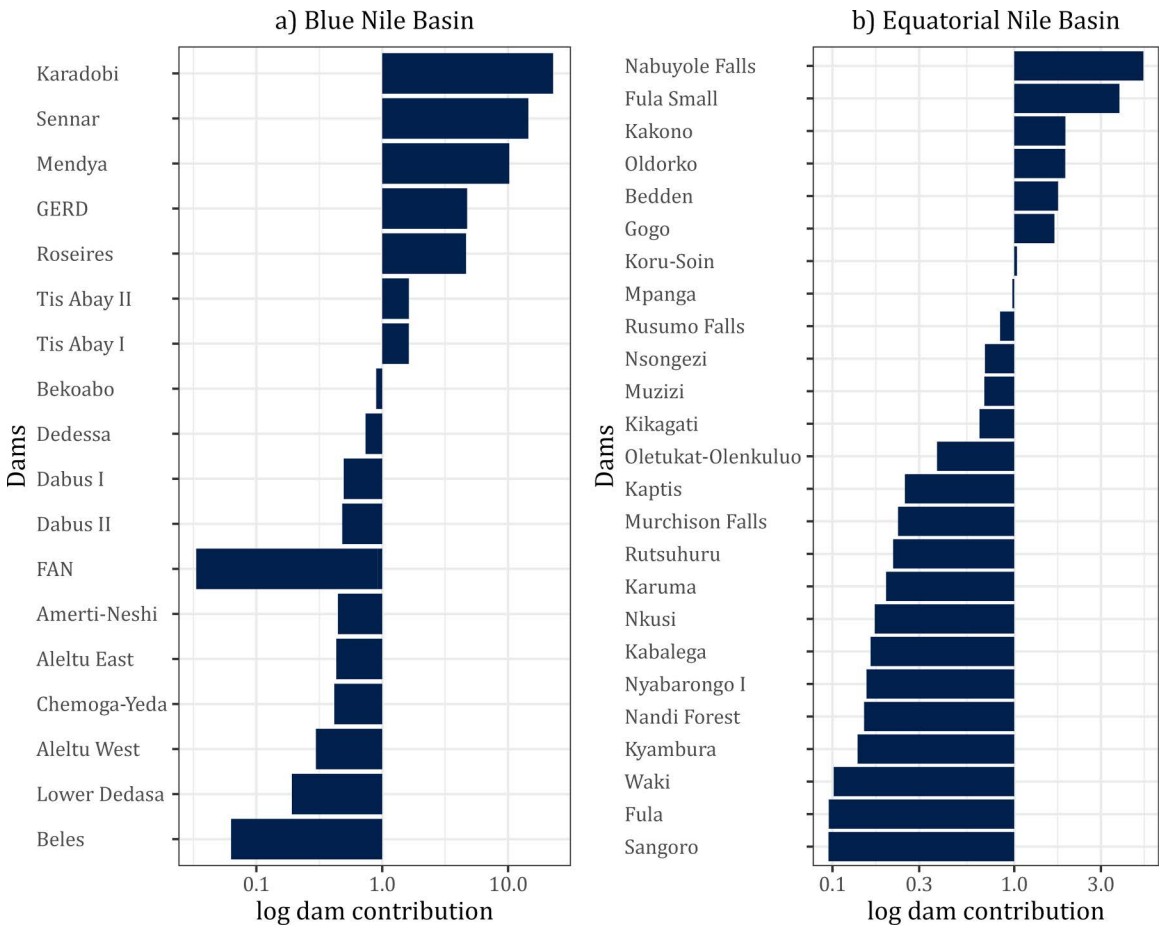

**Fig 8. Log transformed dam contribution to fragmentation of the river network.** (a) The dam contribution uses the leave-on-out principle in the Blue Nile Basin. (b) dam contribution using the leave-one-out principle in the Equatorial Nile Basin (ENB).

We compared our findings with the connectivity assessment in China's Yangtze and Yellow River Basins, where connectivity decreased by 67% and 44%, respectively, from 1980 to 2010 [48]. In our study, if we consider only existing dams (mean RCI up to 2025) and impassable dams, the connectivity decreased by 38% and 58.5% in the BNB and ENB, respectively. The BNB rates were still lower than in the Yangtze River Basin but higher in the ENB. Also, Rodeles et al. (2020) studied eight basins from the Iberian Peninsula, including Spain and Portugal, where the connectivity indices ranged from 3.5% to 20%. The connectivity was higher than in our study basins, even for impassable dam scenarios. The eight basins had more dams per unit surface area of the river basin; for example, the Duero, with a DCI of 3.5%, covers a drainage basin of 91,179 km$^2$ containing 158 dams [7] compared to ~ 300,000 km$^2$ for BNB with 19 existing and proposed dams [40]. We also compared the connectivity of the Big Brook River Basin in Terra Nova National Park (Newfoundland, Canada), which had a connectivity of 29.7% [47]. Although the connectivity for both basins decreased, the current and predicted connectivity is still high compared to other basins. In contrast to our study, we considered low stream (capillary stream orders). Low-stream orders usually contribute 70% of stream network length [58], and they are relatively longer than high-stream orders (Fig S1) [59]. Therefore, discarding them would possibly oversimplify the river network, which reduces the connectivity index as fewer river segments are considered.

## 4.2. Fish-ecological relevance of connectivity in the subbasins

The hydropower dams, especially in the ENB, will likely escalate the species extirpation. The dams have intensely fragmented particular river sections, especially the Upper Victoria Nile (UVN), harboring the threatened and endemic *Neochromis simotes* [60]. Worse still, most species are not yet described [60] and may unknowingly be extirpated from the river catchment. So far, 62 haplochromine cichlids have been identified in the UVN [60]. Other sections of the ENB, like the rivers Kagera and Sio, which also have high hydropower potential, are known habitats for *Labeo victorianus*, which is also restricted to the Victoria basin within the ENB [61,62]. The species has shown separate lacustrine and riverine genetically isolated subpopulations [63]. Dams have led to the inhibition of fish gene flow, leading to the creation of subpopulation; for example, Nalubale dam has led to the inhibition of gene flow for Nile tilapia between lakes Kyoga and Victoria, leading to riverine subpopulation [15]. Further, the Bujagali dam, which became operational in 2012, has altered the ecological setup of the river [57]. The proposed construction of dams, mostly Oletukat Olenkuluo, Leshoto, Amala High, and Oldorko on Mara River in Kenya, are likely to reduce water levels in Lake Victoria Mara Bay, Maisori Swamp, and Serengeti National Park key biodiversity areas [64,65]. Therefore, basin-wide conservation planning should include different ecosystem components, such as fish, that depend on the connectedness of the river networks. Also, ENB should be prioritized for management actions, including an inventory of the existing barriers and identifying obsolete barriers that can be removed.

In the BNB, although the effects of hydropower dams are scanty, irrigation dams and weirs have led to the disruption of migration routes of mostly the cyprinids fish species, such as *Labeobarbus macrophtalmus* [66–68]. Further, predictions in the operation of GERD have shown significant changes in the physiochemical parameters such as turbidity, dissolved oxygen, and nutrient enrichment in the reservoir [69]. Therefore, compounding these stressors without prior plans on examining the migration or survival will lead to the extinction of fish species, mainly the species endemic in the BNB.

## 4.3. Spatial evaluation of the network connectivity using network centrality measures

The siting of hydropower dams is variable within the river network due to hydromorphological factors such as discharge, river slope, and width [70]. In our study, the southwestern and central-eastern regions of the BNB were the most fragmented. For the ENB, the southwestern and central regions were most fragmented. Both basins have high discharge in these regions [33]. For example, the BNB, Ethiopian highlands, and Lake Tana provide high volumes of water to support hydropower production. However, these regions are associated with high species diversity and endemism [68]. Moreover, a general reduction in fish catches is observed downstream of dams, mainly in the central region of the ENB [57].

A strong positive correlation between mean RCI and betweenness centrality underlined the importance of the river reach location to the network fragmentation. Therefore, river segments that were highly connected and fragmented had a higher contribution to fragmentation. [48] had a similar observation in the Yangtze River, where the Three Gorges Dam had less impact on the river fragmentation than the Gezhouba Dam, which is much smaller but located in the middle of the river network [71]. Also, there was a high loss of connectivity in the middle section of the Mekong basin in China by dams, regardless of their size. However, based on [51], the spatial variability also depends on the connectivity index used. For instance, the symmetric RCI was higher for downstream river segments, while asymmetrical RCI was higher in Spain's mid to upstream segments of the Ebro River network [51]. Also, [47] argued that the dam location variably affects migratory species; for example, dams near the oceans significantly affect diadromous species, and those near river inflows mainly affect potamodromous species.

## 4.4. Prioritizing dams for effective management or removal

Removing barriers if they are obsolete, financially unviable to maintain, unsafe, or have significant ecological impacts is the straightforward way to defragment rivers [32]. However, the scarcity of financial resources makes prioritizing barrier removal indispensable [41]. Dams in both basins are not obsolete based on 100 years of operation or longer if the dam

is well managed [10,72]. The oldest dam in the BNB was constructed in 1925 primarily for irrigation but later modified for hydropower production. At present, the Sennar Dam had the highest contribution to network fragmentation of the BNB, followed by GERD. Since the Sennar dam is about 99 years old, efforts should be geared towards its removal to defragment the river basin. Because GERD removal or re-positioning is unlikely, management efforts such as extensive and well-monitored fish migratory corridors should be constructed to maintain species migration to safeguard the *Labeobarbus* species. Regarding the ENB, the Kakono dam had the highest contribution to fragmentation. The dam is still under construction [40], which provides options for mitigation measures, such as constructing a fish pass to increase its dam passability for migratory fishes. The dam is being constructed along River Kagera, the main migratory corridor for *Labeo victorianus* [73]. When all dams (existing, proposed, and under construction) are considered, the highest river fragmentation was contributed by the Karodobi Dam in the BNB and the Nabuyole Falls Dam in the ENB, respectively. Both dams are currently proposed dams [40]. For instance, Nabuyole Falls Dam is a 30-megawatt dam of 54 meters in height proposed on River Nzoia [74]. River Nzoia also drains into Lake Victoria and is a known habitat for an endemic migratory fish species [75]. Therefore, its fragmentation without considering the upstream and downstream migration will extirpate the species from the river basin. Thus, these proposed dams should either not be constructed to avoid further fragmentation of longitudinal connectivity or a comprehensive environmental impact assessment should be carried out that sets out and prescribes clear mitigation measures. Their passability should be assessed to effectively evaluate how these dams contribute to the fragmentation of the river network. Barrier removal is also a complex issue hinged on socioeconomics, ecological and technical aspects, and safety [41,76]. For instance, barriers often slow the invasion of alien species in the ecosystem [77,78]. Thus, a multicriteria approach should be used to evaluate the socioeconomic and ecological tradeoffs while identifying barriers suitable for removal [41,79]. Several barrier removal optimization criteria and tools have been developed to balance the economic and ecological costs and gains [80–82]. However, no approach fits all barriers that should be removed. In our study, we provided a baseline to identify ecologically unsustainable dams to be prioritized for management based on their location, but neither the costs of removing them, the likelihood of invasive species spread, nor the sociocultural dividends of the barriers to the community. Therefore, we do not conclusively identify a dam for removal.

Fish passage facilities were incorporated for existing dams and dams under construction, for example, Karuma Dam [83], Kakono Dam [84], and Kikagati [85]. However, scanty information on the migration patterns of the fishes within the basin has led to difficulties in designing effective fish passage facilities [84]. Fish migration studies should, therefore, be conducted to inform in designing effective fish passes. In the relatively well-studied region (the Upper Victoria Nile) in the ENB, a fish passage was deemed irrelevant for the Bujagali dam, a 250 MW dam [86].

Further, environmental flow (e-flow) analysis is still scanty in comprehensively examining the effects of dams on aquatic biota. So far in the ENB, e-flow has been conducted on Waki small hydropower, and a *no or minimal operator control and no possibility of device failure* principle was a requirement to the dam developer [87]. This unregulated and uninterrupted e-flow release has been successful on Waki SHP, commissioned in 2018, and highly recommended for SHP river valley arrangements [87]. The broad-crest weir was replaced with an interrupted and unregulated pipe from the bottom of the intake chamber [87]. Also, involving local citizens in decision-making concerning water resource management should be enhanced to motivate local people's willingness to pay for reconnecting rivers [88].

## 5. Conclusion

We evaluated the impacts of hydropower dams on the network connectivity in the BNB and the ENB, two sub-basins of the Nile River. We also identified dams to be prioritized for the improved management of the river network connectivity. The connectivity of the ENB has significantly reduced and is below the 50% threshold, even when a high dam passability scenario is assumed. Therefore, constructing the proposed dams, mainly Nabuyole Falls Dam, should be abandoned to avoid a significant decrease in the longitudinal connectivity of ENB. For already constructed dams, if dam removal is not possible, such as GERD in BNB and Kakono in ENB, effective fish passes for migratory fish species should be built to

improve longitudinal connectivity of the sub-basins. One limitation of this study is the potential inaccuracies in dam location data; thus, a systematic inventory of barriers should be created for both basins to support the development of river basin management plans [5]. This study is timely, and we aim to create awareness by including basin-wide river assessments to reduce the environmental impacts of newly built dams and improve the management of existing ones. An open-access database for barriers has been created for Europe [31]. We also recommend a similar approach to establish a database for barriers to comprehensively examine the extent of the river network fragmentation in the basin and Africa. Future research should focus on developing detailed fish migration models to validate the effectiveness of proposed fish passes and migratory corridors. We also recommend assessing the barrier passabilities to effectively determine the extent of longitudinal fragmentation of the river networks. Lastly, a multicriteria approach for assessing dam removal should be conducted.

## Supporting information

**Fig S1. Relationship between stream orders and length for both basins in the Nile Basin.**
(TIF)

## Author contributions

**Conceptualization:** Anthony Basooma, Thomas Hein, Florian Borgwardt.

**Formal analysis:** Anthony Basooma.

**Funding acquisition:** Astrid Schmidt-Kloiber, Thomas Hein, Florian Borgwardt.

**Investigation:** Anthony Basooma, Thomas Hein.

**Methodology:** Anthony Basooma, Rose Basooma, Johannes Kowal, Andrea Funk, Florian Borgwardt.

**Project administration:** Thomas Hein, Florian Borgwardt.

**Resources:** Thomas Hein, Florian Borgwardt.

**Supervision:** Thomas Hein, Florian Borgwardt.

**Validation:** Anthony Basooma, Rose Basooma, Herbert Nakiyende, Johannes Kowal, Andrea Funk.

**Visualization:** Anthony Basooma.

**Writing – original draft:** Anthony Basooma.

**Writing – review & editing:** Anthony Basooma, Astrid Schmidt-Kloiber, Rose Basooma, Herbert Nakiyende, Johannes Kowal, Andrea Funk, Thomas Hein, Florian Borgwardt.

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
