## [Decision Letter · Decision Letter 0]

6 Aug 2024

PONE-D-24-20020Spatiotemporal changes in river network connectivity in the Nile River Basin due to hydropower damsPLOS ONE

Dear Dr. Basooma,

Thank you for submitting your manuscript to PLOS ONE. After careful consideration, we feel that it has merit but does not fully meet PLOS ONE’s publication criteria as it currently stands. Therefore, we invite you to submit a revised version of the manuscript that addresses the points raised during the review process.

We look forward to receiving your revised manuscript.

Kind regards,

AL MAHFOODH

Academic Editor

PLOS ONE

Journal Requirements:

"Anthony Basooma acknowledges funding from AquaINFRA (agreement ID: 101094434) and DANUBE4All (grant agreement no. 101093985) projects funded by the European Commission. The financial support by the HR21 Doctoral School of the University of Natural Resources and Life Sciences, Vienna, Austria, the Austrian Federal Ministry for Digital and Economic Affairs, the National Foundation for Research, Technology, and Development, and the Christian Doppler Research Association is gratefully acknowledged."

5. We note that [Figures 1,5 and 6] in your submission contain [map/satellite] images which may be copyrighted. All PLOS content is published under the Creative Commons Attribution License (CC BY 4.0), which means that the manuscript, images, and Supporting Information files will be freely available online, and any third party is permitted to access, download, copy, distribute, and use these materials in any way, even commercially, with proper attribution. For these reasons, we cannot publish previously copyrighted maps or satellite images created using proprietary data, such as Google software (Google Maps, Street View, and Earth). For more information, see our copyright guidelines: http://journals.plos.org/plosone/s/licenses-and-copyright.

a. You may seek permission from the original copyright holder of Figures 1,5 and 6 to publish the content specifically under the CC BY 4.0 license.  

Reviewers' comments:

Reviewer's Responses to Questions

**Comments to the Author**

1. Is the manuscript technically sound, and do the data support the conclusions?

Reviewer #1: Yes

Reviewer #2: Partly

2. Has the statistical analysis been performed appropriately and rigorously? 

Reviewer #1: Yes

Reviewer #2: Yes

3. Have the authors made all data underlying the findings in their manuscript fully available?

Reviewer #1: No

Reviewer #2: Yes

4. Is the manuscript presented in an intelligible fashion and written in standard English?

Reviewer #1: Yes

Reviewer #2: Yes

5. Review Comments to the Author

Reviewer #1: General considerations:

The authors assessed the impacts of hydropower dams on the network connectivity of two Nile River sub-basins. Additionally, they try to identify the dams to be prioritized to improve river network connectivity.

This work is rather ambitious since it deals with a large and complex area, with high fragmentation. Besides, the period considered in the evaluation of the progressive loss of connectivity is quite long. For this purpose, the authors used a common connectivity index (RCI) and network centrality measures to assess temporal and spatial variation in longitudinal connectivity. This seems to be an adequate methodology and the work is interesting and with innovative contribution.

However, the authors seem to have considered only hydropower plants, whereas other types of transversal structures, namely for irrigation, or water supply, were not included. This option was not conveniently explained. I also think that Information about the loss of biodiversity and protected habitats is very scarce and it should be more developed.

Conclusions are more or less obvious respecting the new proposed dams since the number of impassible barriers is already very high. It would be much more convenient for the authors to focus their work on the dams already built where they could be installed (or improved) the fish passes. But this is an important gap (the authors just assigned downstream and upstream passability values, but far from reality), this is, the absence of information about the observed passability of each dam or the existence of environmental flows. Probably, this information is difficult to obtain (the authors state that is unavailable), but it should be considered in the future towards the management process (and mitigation). Probably these aspects are more realistic in terms of catchment management than barrier removal _especially the big structures, like the authors conclude… Anyway, the results obtained about the individual contribution of each dam are useful if we consider the focus on the passability improvement in those sections, which do not implicate necessarily their removal. .Of course, the authors only consider connectivity, mainly directed to fish populations, but other social and economic criteria should be included as well (many authors consider multicriteria models with this objective). Please, consider these aspects for the discussion to become more complete.

Moreover, the 1st hypothesis is obvious and it should be discarded an increase in the number of dams constructed along the river network has significantly reduced the longitudinal connectivity of both basins over time).

I believe that the 1st part of 4.1 could be moved to the introduction since it deals with the historical temporal trends in hydropower dam construction.

Other aspects:

How far it was important to include low-stream orders? Please mention it in the discussion.

Line 152: Please explain the acronyms li and L in the formula.

Line 198. Explain better what is RCIstart.

Line 205 This correlation was obtained between which type of variables?

Line 288. Which period?

Reviewer #2: Writing Quality

Page 1, Abstract: The abstract is wide, and lacks focus on key findings

Line 46: Unnecessary use of "river networks”. Suggested: “The growing demand for energy and flood control has led to increased construction of barriers along rivers."

Line 55-56: Redundancy and lack of specificity.

One-time abbreviations of the BNB and NRB at the beginning are enough.

Methods

Line 112-118: Lack of detail on data sources and processing.

Line 113-119: Insufficient detail on dam data collection. Also figures at the end have not numbers as captions.

Line 135: Lack of detail on the alignment of dams to river networks. (in Arcmap? Which version)

Line 152-160: Incomplete explanation for variables in the equations 2.1-2.3

Results

Repetitive use of the word “passability”

Line 204- 205: Statement on dam numbers is not clear.

Line 208: Lack of emphasis on significance of results. You can say “scenarios indicate a significant decrease in RCI….”

Line 243 Figure 2 Caption: Insufficient detail in figure caption. You can explain more the figure in the caption

Discussion

The discussion section provides a good interpretation but lacks depth in addressing limitations and actionable recommendations.

Line 296-298: Repetitive and lacks depth. Suggestion: "We thoroughly examined the temporal and spatial patterns in the river network connectivity of the Equatorial Nile and Blue Nile, focusing on individual dam contributions to overall river network fragmentation."

Line 340-346: you can add: "One limitation of this study is the potential inaccuracies in dam location data, which were cross verified using available literature and datasets."

Figures

The figures are located at the end of the manuscript and are well-designed but need higher resolution and detailed captions.

For example, for Figure 2:

Improve resolution: Detailed suggested Caption: "Figure 2. Trends in dam construction from 1925 to 2035 for the Blue Nile Basin (BNB) and Equatorial Nile Basin (ENB). The figure illustrates the increase in dam numbers over time, with a noticeable surge post-2010."

Figure 5 and 6 the grids latitude and longitude are not readable.

Also yellow is not a good choice for the Figure 7.

Science Question

The research question is well-defined but needs explicit discussion on how the results validate the hypotheses stated.

At the end of discussion or In conclusion it can be added: "Future research should focus on developing detailed fish migration models to validate the effectiveness of proposed fish passes and migratory corridors."

Literature

Missing recent studies: Incorporate more recent studies to (after 2022) provide an updated perspective on the impacts of dam construction on river connectivity.

6. PLOS authors have the option to publish the peer review history of their article (what does this mean? ). If published, this will include your full peer review and any attached files.

**Do you want your identity to be public for this peer review?** For information about this choice, including consent withdrawal, please see our Privacy Policy .

Reviewer #1: **Yes: ** Rui Manuel Vitor Cortes

Reviewer #2: No

---

## [Author Response · Author response to Decision Letter 1]

29 Oct 2024

Response to reviewers

We are very grateful for the tremendous effort by the reviewers and the editor for the comments provided to improve our work. We have accordingly addressed the comments and hope the study will be impactful but, most importantly, aid in managing the Nile River Basin in general.

Editor comments

Response: the manuscript has been fully updated based on the required formatting

Response: Since the study was not field-based, we did not require any research permit for this work.

Thank you for stating the following financial disclosure:

"Anthony Basooma acknowledges funding from AquaINFRA (agreement ID: 101094434) and DANUBE4All (grant agreement no. 101093985) projects funded by the European Commission. The financial support by the HR21 Doctoral School of the University of Natural Resources and Life Sciences, Vienna, Austria, the Austrian Federal Ministry for Digital and Economic Affairs, the National Foundation for Research, Technology, and Development, and the Christian Doppler Research Association is gratefully acknowledged."

Response: The funding information has been updated in the system, and the manuscript is on lines 458 to 463.

Response: the statement “The funders had no role in study design, data collection and analysis, decision to publish, or preparation of the manuscript” has been incorporated on lines 464 and 465.

5. We note that [Figures 1,5 and 6] in your submission contain [map/satellite] images which may be copyrighted. All PLOS content is published under the Creative Commons Attribution License (CC BY 4.0), which means that the manuscript, images, and Supporting Information files will be freely available online, and any third party is permitted to access, download, copy, distribute, and use these materials in any way, even commercially, with proper attribution. For these reasons, we cannot publish previously copyrighted maps or satellite images created using proprietary data, such as Google software (Google Maps, Street View, and Earth). For more information, see our copyright guidelines: http://journals.plos.org/plosone/s/licenses-and-copyright.

a. You may seek permission from the original copyright holder of Figures 1,5 and 6 to publish the content specifically under the CC BY 4.0 license.

Response: Figures 1, 5, and 6 were regenerated from HydroATLAS, which forms a Collective Database, i.e., a collection of information from independent datasets, and as a whole is licensed under a Creative Commons Attribution 4.0 International License (CC-BY 4.0) (https://data.hydrosheds.org/file/technical-documentation/HydroATLAS_TechDoc_v10.pdf).

Reviewers

Comments to the Author

1. Is the manuscript technically sound, and do the data support the conclusions?

Reviewer #1: Yes

Reviewer #2: Partly

2. Has the statistical analysis been performed appropriately and rigorously?

Reviewer #1: Yes

Reviewer #2: Yes

3. Have the authors made all data underlying the findings in their manuscript fully available?

Reviewer #1: No

Response: We have provided links for all the data used and supplementary table S1 for the dataset for the dams. We created a Figshare account to archive the river network data and develop the maps on lines 480 to 485.

Reviewer #2: Yes

4. Is the manuscript presented in an intelligible fashion and written in standard English?

Reviewer #1: Yes

Reviewer #2: Yes

5. Review Comments to the Author

Reviewer #1: General considerations:

The authors assessed the impacts of hydropower dams on the network connectivity of two Nile River sub-basins. Additionally, they try to identify the dams to be prioritized to improve river network connectivity.

This work is rather ambitious since it deals with a large and complex area, with high fragmentation. Besides, the period considered in the evaluation of the progressive loss of connectivity is quite long. For this purpose, the authors used a common connectivity index (RCI) and network centrality measures to assess temporal and spatial variation in longitudinal connectivity. This seems to be an adequate methodology and the work is interesting and with innovative contribution.

However, the authors seem to have considered only hydropower plants, whereas other types of transversal structures, namely for irrigation, or water supply, were not included. This option was not conveniently explained.

Response: Due to data availability, the current study focused on hydropower plants, especially confirming the dam's location and the commissioning time. Unfortunately, obtaining the necessary information for small barriers was difficult. This was provided on lines 137 to 138.

I also think that Information about the loss of biodiversity and protected habitats is very scarce and it should be more developed.

Response: we have created a section in the discussion to briefly highlight the likely and already existing impacts on biodiversity, mostly fish, due to dams in both basins on lines 352 to 377.

Conclusions are more or less obvious respecting the new proposed dams since the number of impassible barriers is already very high. It would be much more convenient for the authors to focus their work on the dams already built where they could be installed (or improved) the fish passes. But this is an important gap (the authors just assigned downstream and upstream passability values, but far from reality), this is, the absence of information about the observed passability of each dam or the existence of environmental flows. Probably, this information is difficult to obtain (the authors state that is unavailable), but it should be considered in the future towards the management process (and mitigation).

Response: We agree with the reviewer and have included the statement recommending further research on fish migration models to validate the available fish passes. Also, the need to evaluate the passability of the barriers for fish needs to be conducted on lines 452 to 456.

Probably these aspects are more realistic in terms of catchment management than barrier removal _especially the big structures, like the authors conclude… Anyway, the results obtained about the individual contribution of each dam are useful if we consider the focus on the passability improvement in those sections, which do not implicate necessarily their removal. Of course, the authors only consider connectivity, mainly directed to fish populations, but other social and economic criteria should be included as well (many authors consider multicriteria models with this objective). Please, consider these aspects for the discussion to become more complete.

Response: we have included a section on the other criteria of barrier removal in the discussion on lines 427 to 436.

Moreover, the 1st hypothesis is obvious and it should be discarded an increase in the number of dams constructed along the river network has significantly reduced the longitudinal connectivity of both basins over time).

Response: We acknowledge that the first hypothesis, regarding the reduction in longitudinal connectivity due to increased dams, may seem obvious. However, we included this hypothesis to establish a foundational understanding of the impact of dam construction over time. This baseline is crucial for contextualizing our subsequent analyses and comparing the extent of connectivity loss across different sections of the river network. We have restructured it on lines 101 to 102

I believe that the 1st part of 4.1 could be moved to the introduction since it deals with the historical temporal trends in hydropower dam construction.

Response: Although the section deals with a historical perspective of dam construction, it was brought up to discuss the temporal change in dam construction and why changes occurred over those times.

Other aspects:

How far it was important to include low-stream orders? Please mention it in the discussion.

Response (lines 347 to 351): A section on including low-stream orders has been incorporated in the discussion, and a supplementary figure, Fig S1 (lines 710 to 712), expounds on the aspect.

Line 152: Please explain the acronyms li and L in the formula.

Response: We have included the meaning of the acronyms on line 160

Line 198. Explain better what is RCIstart.

Response: we renamed the RCIstart to CCIstart to indicate the initial catchment when all barriers are considered on the river network noted on line 204.

Line 205 This correlation was obtained between which type of variables?

Response: the statement has been removed and maintained as a description of the steady dam increase over time.

Line 288. Which period?

Response:

REVIEWER #2: WRITING QUALITY

Page 1, Abstract: The abstract is wide, and lacks focus on key findings

Response: We have revised the abstract to provide a clearer and more focused summary of the key findings. We believe these changes enhance the clarity and impact of the abstract, and we hope it now better reflects the core contributions of our study.

Line 46: Unnecessary use of "river networks”. Suggested: “The growing demand for energy and flood control has led to increased construction of barriers along rivers."

Response: We agree with the reviewers and changed the text accordingly on line 50

Line 55-56: Redundancy and lack of specificity.

Response: removed

One-time abbreviations of the BNB and NRB at the beginning are enough.

Response: We agree with the reviewer and changed the abbreviation with the full names left in Figure captions and first mentions.

Methods

Line 112-118: Lack of detail on data sources and processing.

Response: This was addressed by providing the reference, data links, and supplementary material for updated data for hydropower dams on lines 112 to 124.

Line 113-119: Insufficient detail on dam data collection. Also figures at the end have not numbers as captions.

Response: The dams used were obtained from the RePP, an open-access dataset. We have archived the extracted reviewed dataset used in this analysis, accessible at https://doi.org/10.6084/m9.figshare.26886409.v1. This has been indicated in lines 120 to 123.

Line 135: Lack of detail on the alignment of dams to river networks. (in Arcmap? Which version)

Response: we used Q-GIS ver 3.28.4 addressed on lines 114.

Line 152-160: Incomplete explanation for variables in the equations 2.1-2.3

Response: the variables have been well described on lines 160 to 165

Results

Repetitive use of the word “passability”

Response: addressed – rephrased the text.

Line 204- 205: Statement on dam numbers is not clear.

Response: the statement has been rephrased on lines 209 to 210.

Line 208: Lack of emphasis on significance of results. You can say “scenarios indicate a significant decrease in RCI….”

Response: The text changed accordingly on lines 212 to 213

Line 2

---

## [Decision Letter · Decision Letter 1]

26 Feb 2025

PONE-D-24-20020R1Spatiotemporal changes in river network connectivity in the Nile River Basin due to hydropower damsPLOS ONE

Dear Dr. Basooma,

Thank you for submitting your manuscript to PLOS ONE. After careful consideration, we feel that it has merit but does not fully meet PLOS ONE’s publication criteria as it currently stands. Therefore, we invite you to submit a revised version of the manuscript that addresses the points raised during the review process.

We look forward to receiving your revised manuscript.

Kind regards,

Halil Ibrahimi

Academic Editor

PLOS ONE

Journal Requirements:

Additional Editor Comments:

Please check comments of the reviewers and proceed with the corrected version of manuscript.

Reviewers' comments:

Reviewer's Responses to Questions

**Comments to the Author**

1. If the authors have adequately addressed your comments raised in a previous round of review and you feel that this manuscript is now acceptable for publication, you may indicate that here to bypass the “Comments to the Author” section, enter your conflict of interest statement in the “Confidential to Editor” section, and submit your "Accept" recommendation.

Reviewer #1: (No Response)

Reviewer #3: (No Response)

2. Is the manuscript technically sound, and do the data support the conclusions?

Reviewer #1: Yes

Reviewer #3: Yes

3. Has the statistical analysis been performed appropriately and rigorously? 

Reviewer #1: Yes

Reviewer #3: Yes

4. Have the authors made all data underlying the findings in their manuscript fully available?

Reviewer #1: Yes

Reviewer #3: Yes

5. Is the manuscript presented in an intelligible fashion and written in standard English?

Reviewer #1: Yes

Reviewer #3: Yes

6. Review Comments to the Author

Reviewer #1: This is, in fact, an ambitious work, covering a vast catchment, facing the difficulties of scarce available (and detailed) data about the transversal structures that impose a loss of connectivity. The hypotheses tested are more or less obvious (as well as the impact of the new structures to be built), especially the 1st hypothesis testing if there is a significant temporal increase in the number of dams constructed …? I think that this hypothesis could be discarded. However, the authors obtained some interesting conclusions, like the importance of the contribution of the dam in the river reach to the global network fragmentation.

I acknowledge that the authors improved substantially the original paper following the indications of both reviewers. However, some questions remain, some of them that were already pointed out by the reviewer(s). For instance, it was identified the dams that were prioritized to improve connectivity. But how to mitigate, for these specific dams the effects of fragmentation? The idea that effective fish passes for migratory fish species should be built represents a nice intention but is that feasible considering the respective dam heights? What are the most convenient fish passes for each case? Were installed in this catchment any fish passes (and defined environmental flows) and is it possible to know their relative success?. I believe that the removal option to improve connectivity is not a real possibility in this catchment (except for small weirs with low relevance in the connectivity). The authors should introduce these aspects in the discussion, following also previous considerations by the reviewers.

Reviewer #3: The authors present a good research paper on the impacts of hydroelectric dams on the connectivity of two sub-basins of the Nile River. In previous revisions, they added the suggestions and comments made by the reviewers, strengthening their study.

My comments are basically directed at two recommendations related to the two most significant weaknesses that the authors make in their conclusions.

Regarding the need to create a systematic inventory of barriers for both basins, he recommended that the authors read, and if they wish, the bibliographical citation of this article.

https://doi.org/10.1038/s41586-020-3005-2

And on the need to raise awareness by including basin-wide river assessments, examples from other parts of the world could be helpful. To this end, I suggest that the authors read, and if they wish, cite the bibliography of these two articles:

https://doi.org/10.1007/s11273-022-09864-6

https://doi.org/10.1038/s41598-022-10170-7

7. PLOS authors have the option to publish the peer review history of their article (what does this mean? ). If published, this will include your full peer review and any attached files.

**Do you want your identity to be public for this peer review?** For information about this choice, including consent withdrawal, please see our Privacy Policy .

Reviewer #1: No

Reviewer #3: No

---

## [Author Response · Author response to Decision Letter 2]

14 Mar 2025

Journal Requirements:

Response: We have critically checked the reference list for spellings, consistencies, and doi validity on lines

Additional Editor Comments:

Please check comments of the reviewers and proceed with the corrected version of manuscript.

Reviewers' comments:

Reviewer's Responses to Questions

Comments to the Author

1. If the authors have adequately addressed your comments raised in a previous round of review and you feel that this manuscript is now acceptable for publication, you may indicate that here to bypass the “Comments to the Author” section, enter your conflict of interest statement in the “Confidential to Editor” section, and submit your "Accept" recommendation.

Reviewer #1: (No Response)

Reviewer #3: (No Response)

2. Is the manuscript technically sound, and do the data support the conclusions?

Reviewer #1: Yes

Reviewer #3: Yes

Response: We thank the reviewers for acknowledging that our manuscript is technically sound and that data supports the conclusions.

3. Has the statistical analysis been performed appropriately and rigorously?

Reviewer #1: Yes

Reviewer #3: Yes

Response: We thank the reviewers for acknowledging that the statistical analysis was appropriate and rigorously conducted.

4. Have the authors made all data underlying the findings in their manuscript fully available?

Reviewer #1: Yes

Reviewer #3: Yes

Response: We thank the reviewers for confirming the presence of underlying data that enables the reproducibility of our study.

5. Is the manuscript presented in an intelligible fashion and written in standard English?

Reviewer #1: Yes

Reviewer #3: Yes

Response: We thank the reviewers for agreeing that our manuscript was written in an intelligible manner.

6. Review Comments to the Author

Reviewer #1: This is, in fact, an ambitious work, covering a vast catchment, facing the difficulties of scarce available (and detailed) data about the transversal structures that impose a loss of connectivity. The hypotheses tested are more or less obvious (as well as the impact of the new structures to be built), especially the 1st hypothesis testing if there is a significant temporal increase in the number of dams constructed …? I think that this hypothesis could be discarded. However, the authors obtained some interesting conclusions, like the importance of the contribution of the dam in the river reach to the global network fragmentation.

Response: we agree with the reviewer that the first objective was quite obvious but vital for the study because it showed that between the two basins, the increase in the number of dams was not the same. For instance, a significant increase was mainly in the Equatorial Nile region.

I acknowledge that the authors improved substantially the original paper following the indications of both reviewers. However, some questions remain, some of them that were already pointed out by the reviewer(s). For instance, it was identified the dams that were prioritized to improve connectivity. But how to mitigate, for these specific dams the effects of fragmentation?

The idea that effective fish passes for migratory fish species should be built represents a nice intention but is that feasible considering the respective dam heights? What are the most convenient fish passes for each case? Were installed in this catchment any fish passes (and defined environmental flows) and is it possible to know their relative success?. I believe that the removal option to improve connectivity is not a real possibility in this catchment (except for small weirs with low relevance in the connectivity). The authors should introduce these aspects in the discussion, following also previous considerations by the reviewers.

Response: we addressed the questions on lines 441 to 455

Reviewer #3: The authors present a good research paper on the impacts of hydroelectric dams on the connectivity of two sub-basins of the Nile River. In previous revisions, they added the suggestions and comments made by the reviewers, strengthening their study.

My comments are basically directed at two recommendations related to the two most significant weaknesses that the authors make in their conclusions.

Regarding the need to create a systematic inventory of barriers for both basins, he recommended that the authors read, and if they wish, the bibliographical citation of this article.

https://doi.org/10.1038/s41586-020-3005-2

Response: Incorporated on L77 and also L470

And on the need to raise awareness by including basin-wide river assessments, examples from other parts of the world could be helpful. To this end, I suggest that the authors read, and if they wish, cite the bibliography of these two articles:

https://doi.org/10.1007/s11273-022-09864-6

https://doi.org/10.1038/s41598-022-10170-7

Response: Included in the lines 41 to 45 as a recommendation for environmental monitoring

7. PLOS authors have the option to publish the peer review history of their article (what does this mean?). If published, this will include your full peer review and any attached files.

Do you want your identity to be public for this peer review? For information about this choice, including consent withdrawal, please see our Privacy Policy.

Reviewer #1: No

Reviewer #3: No

---

## [Editor Report · Decision Letter 2]

21 Mar 2025

Spatiotemporal changes in river network connectivity in the Nile River Basin due to hydropower dams

PONE-D-24-20020R2

Dear Dr. Basooma,

We’re pleased to inform you that your manuscript has been judged scientifically suitable for publication and will be formally accepted for publication once it meets all outstanding technical requirements.

Kind regards,

Halil Ibrahimi

Academic Editor

PLOS ONE
---

## [Editor Report · Acceptance letter]

PONE-D-24-20020R2

PLOS ONE

Dear Dr. Basooma,

I'm pleased to inform you that your manuscript has been deemed suitable for publication in PLOS ONE. Congratulations! Your manuscript is now being handed over to our production team.

Kind regards,

on behalf of

Professor Halil Ibrahimi

Academic Editor

PLOS ONE